# Effects of Large Extracellular Vesicles from Kidney Cancer Patients on the Growth and Environment of Renal Cell Carcinoma Xenografts in a Mouse Model

Matthieu Ferragu [1],*, Luisa Vergori [2], Vincent Le Corre [1], Sarah Bellal [3], Maria del Carmen Martinez [2],† and Pierre Bigot [1],*

1 Urology Department, Angers University Hospital, 49100 Angers, France
2 INSERM Unite Mixte de Recherche (UMR) 1063, Stress Oxydant et Pathologies Metaboliques, 49100 Angers, France
3 Anatomopathological Department, Angers University Hospital, 49100 Angers, France
* Correspondence: matthieu.ferragu@chu-angers.fr (M.F.); pibigot@chu-angers.fr (P.B.)
† Present address: PhyMedExp, INSERM, CNRS UMR, University of Montpellier, 34090 Montpellier, France.

**Abstract:** Plasma membrane-derived vesicles, also referred to as large extracellular vesicles (lEVs), are implicated in several pathophysiological situations, including cancer. However, to date, no studies have evaluated the effects of lEVs isolated from patients with renal cancer on the development of their tumors. In this study, we investigated the effects of three types of lEVs on the growth and peritumoral environment of xenograft clear cell renal cell carcinoma in a mouse model. Xenograft cancer cells were derived from patients' nephrectomy specimens. Three types of lEVs were obtained from pre-nephrectomy patient blood (cEV), the supernatant of primary cancer cell culture (sEV) and from blood from individuals with no medical history of cancer (iEV). Xenograft volume was measured after nine weeks of growth. Xenografts were then removed, and the expression of CD31 and Ki67 were evaluated. We also measured the expression of MMP2 and Ca9 in the native mouse kidney. lEVs from kidney cancer patients (cEV and sEV) tend to increase the size of xenografts, a factor that is related to an increase in vascularization and tumor cell proliferation. cEV also altered organs that were distant from the xenograft. These results suggest that lEVs in cancer patients are involved in both tumor growth and cancer progression.

**Keywords:** cancer; renal cancer; large extracellular vesicle; tumor growth; angiogenesis; peritumoral environment; xenograft; mouse model

## 1. Introduction

Kidney cancer is an insidious disease, accounting for approximately 2% of cancer diagnoses and deaths worldwide. In 2020, 431,288 new cases and 179,368 deaths from this cancer were reported across the world [1]. There are several histological types of kidney cancer. The most common subtype is clear cell renal cell carcinoma (ccRCC), which accounts for 75% of all cases [2]. Localized kidney cancer has a good prognosis, in which surgery is the only curative treatment, and the 5-year survival is over 80% [3–5]. However, when the cancer progresses and becomes metastatic, the median survival drops to 55.7 months [6]. Tumor progression and metastasis have a strong impact on patient survival and the quality of life of kidney cancer patients [7,8]. The mechanisms by which tumors grow and disseminate in the body are not completely elucidated, but large extracellular vesicles (lEVs) may play a key role.

lEVs are submembrane fragments released into the extracellular environment and involved in intercellular communication [9]. Their biogenesis results from a contraction phenomenon of the cell cytoskeleton after an increase in cytoplasmic calcium [10,11]. A budding is formed on the surface of the cell membrane, following which the lEVs are

released into the extracellular space and circulate in the various fluids of the body such as blood, urine, saliva and breast milk [12]. Released lEVs dock with recipient cells to transmit information from the cell surface or to transfer their contents into the cells. Effects that can be induced by lEVs depend on their membrane and cytosolic proteins and lipids, as well as nucleic acids (DNA, messenger RNA, microRNA, exoRNA), and therefore on their cell of origin [13–15]. All cells produce lEVs, and they are involved in several pathophysiological situations. Cancer cells produce large amounts of lEVs that can interact locally or remotely with target cells via the bloodstream. Several studies have highlighted the role of lEVs in cancer oncogenesis. On the one hand, lEVs act directly on tumor cells in an autocrine manner. For example, glioblastoma-derived lEVs promoted the proliferation of human glioma cell lines in an in vitro model [16]. The paracrine action of lEVs on cancer cells has also been shown in renal cancer. lEVs released from human Wharton's jelly mesenchymal stem cells (hWJ-MSCs) have shown an increase in the growth of kidney cancer cells (786-0) in a mouse model [17]. These results were related to an increase in hepatocyte growth factor (HGF) expression and to the activation of AKT and ERK1/2 signaling pathways in target cells. These lEVs also demonstrated in vitro an increase in the migration and invasion capacities of renal cells with an increase in the expression of chemokine receptor type 4 (CXCR4) and matrix metalloproteinase (MMP-9) [17]. In addition, lEVs alter cancer-associated cells in the peritumoral environment, such as stromal cells. Previous research has shown that MSCs stimulated with lEVs derived from kidney cancer cells have a protumorigenic phenotype. In vitro, these lEV-stimulated MSCs were characterized by a high expression of proteins associated with cell migration (CXCR4) and matrix remodeling (MMP, COL4A3) [18]. In vivo analyses confirmed that these lEV-stimulated MSCs promoted tumor growth and vascularization [18]. lEVs also modify peritumoral angiogenesis. One study showed that lEVs derived from 786-0 kidney cancer cells significantly promote tubular formation in human umbilical vein endothelial cells (HUVECs) via upregulation of vascular endothelial growth factor (VEGF) expression [19]. Finally, lEVs also act at a distance from the tumor to form pre-metastatic niches [20]. Thus, a cancer cell can create new metastases by migrating into an environment already favorable to its development. There are a number of mechanisms by which lEVs interfere in the development, progression and dissemination of cancers and in particular kidney cancer.

Most research to date, however, has studied the effect of lEVs produced in vitro on immortal cell lines. These results may not be representative of the interaction between lEVs and kidney cancer in patients. Until now, to the best of our knowledge, no study has evaluated the effects of lEVs isolated from renal cancer patients on the development of their tumors in an in vivo model.

Therefore, in the present study, we developed a novel model that investigates the pro-tumorigenic action of patients' lEVs on their own tumor, by performing a xenograft in a mouse model. Five patients with operable kidney cancer were selected, and their lEVs and culture cancer cells from the nephrectomy specimen were isolated. The primary cell lines obtained were injected into the flank of nude balb/c mice, and the lEVs were used to treat the mice before, during and after the xenograft was performed. The effects of the patient's lEVs on the xenograft volume and the modification of the peritumoral environment were analyzed.

## 2. Materials and Methods

### 2.1. Patients Included

Five patients with ccRCC were included in this study to obtain kidney specimens and blood samples. These patients were treated by nephrectomy for renal cancer in our center between 15 January and 19 May 2021. The only exclusion criterion was the presence of another cancer. Patient characteristics are reported in Table 1. All patients were from the national database "UroCCR" (NCT03293563, CNIL authorization number: DR-2013-206) and were informed about the purpose of the study, and their written consent was obtained. Authorization from the committee for personal protection was obtained. Control lEVs (iEV)

were isolated from 5 individual donors (41 to 61 years old) with no known medical history of cancer.

**Table 1.** Characteristics of the five patients from whom the primary cultures were obtained. T stage refers to the American Joint Committee on Cancer (AJCC) "Tumor Node Metastasis" (TNM) classification [21]. T describes the size of the original (primary) tumor. N describes lymph nodes that are invaded. M describes distant metastasis. ISUP refers to the International Society of Urological Pathology score.

|  | **5 Patients** |
|---|---|
| Gender |  |
| • Female | 2 |
| • Male | 3 |
| Medianage (Year, SD) | 63.8 (12.4) |
| TStage |  |
| • T1 | 1 |
| • T2 | 2 |
| • T3 | 2 |
| • T4 | 0 |
| • N1 | 0 |
| • M1 | 1 |
| Mediantumorsize (cm, SD) | 8.6 (1.7) |
| ISUPscore |  |
| • 1–2 | 2 |
| • 3–4 | 3 |
| Tumoralnecrosis | 2 |

### 2.2. Primary Cell Isolation and Culture

Once the nephrectomy was performed, the surgical specimen was transported in a fresh buffered salt solution to the anatomopathological department for macroscopic examination. After anatomopathological analysis, a 1 cm$^3$ non-necrotic fragment of the tumor was selected and stored at 4 °C in 50 mL of PBS (phosphate-buffered saline, Sigma Aldrich, St. Louis, MO, USA) until its use. Following this, mechanical digestion was performed under sterile conditions using two cold scalpels. The resulting millimeter-sized pieces were incubated at 37 °C for 90 min under agitation in the presence of DMEM 4.5 g/L glucose, 1% pyruvate, 1% penicillin–streptomycin (Sigma Aldrich), 0.2 Wünsch units/mL liberase (Roche, Basel, Switzerland) and 0.1 mg/mL DNase (Roche). The solution obtained was sieved through a 70 µm filter (SPL Life Sciences, Pocheon, Republic of Korea). Suspended cells were pelleted for 5 min at 300× *g*. Next, cells were subcultured in culture medium 1 containing 33% of Mix AmnioMAX™ (AmnioMAX™medium + AmnioMAX™ supplement) (Thermo Fisher Scientific, Waltham, MA, USA), 10% of Fetal Bovine Serum (FBS) (Thermo Fisher Scientific), 1% of RGEM1™ (LONZA, Basel, Switzerland), 1% of Glutamine (Sigma-Aldrich), 1% of penicillin and streptomycin (Sigma-Aldrich) and 54% of DMEM 4.5% of glucose (Sigma-Aldrich) prewarmed to 37 °C and transferred to 25 cm$^2$ culture flasks (SPL Life Sciences). After overnight incubation (37 °C, 5% CO$_2$), culture medium was renewed, and the unattached cells were removed. This procedure was repeated every 24 h until the flasks were free of debris and unattached cells. Thereafter, the culture medium was changed every 2 to 3 days until confluence. At 70–80% confluence, cells were washed with PBS and detached using trypsin (20 µL/cm$^2$ trypsin-EDTA 1X (trypsin 0.05%; EDTA 0.02%), Sigma-Aldrich) for 5 min. Cells were then pelleted by 5 min centrifugation at 500× *g* and resuspended in 2 mL of culture medium 2 containing 33% of Mix AmnioMAX™ (Thermo Fisher Scientific), 10% of FBS (Thermo Fisher Scientific), 1% of Glutamine (Sigma-Aldrich), 56% of DMEM and 4.5% of glucose (Sigma-Aldrich). Cell

count was performed by trypan blue exclusion and Neubauer hemocytometer. Some of the cells were frozen at $-80\ °C$, and the rest were transferred to a 75 $cm^2$ flask with culture medium 2. Once 80% confluence was obtained, the cells were transferred to a T175 $cm^2$ flask for amplification. Culture medium 1 was optimized for survival after mechanical and chemical digestion and to avoid culture infections during tissue handling. It contained growth factor supplements for use with renal epithelial cell basal medium (1% REGM1™ (Renal Epithelial Cell Growth Medium SingleQuots™ Kit, LONZA, Bâle, Switzerland)) and 1% antibiotic (penicillin and streptomycin, Sigma-Aldrich). After the first passage, these elements were no longer needed and could constitute an avoidable bias for cell and xenograft growth. They were removed from culture medium 2.

### 2.3. Immunofluorescence

Primary cultures were characterized by immunofluorescence by labeling different renal and renal cancer markers such as CA9 (Cell Signaling, Technology, Danvers, MA, USA), Pax8, CD10, MIF1a and fibroblast activation protein alpha (FAP-alpha) (all Abcam, Cambridge, MA, USA). For this, $10^6$ cells were plated on a sterilized 12-well glass slide and then allowed to grow in culture medium 2. Once the cells had reached 80% confluence, they were fixed with paraformaldehyde (2%, Thermo Fisher Scientific) at room temperature for 5 min and washed. After saturating with 5% bovine serum albumin (BSA) (Sigma-Aldrich) in PBS for 30 min at room temperature, the cells were incubated with the primary antibody overnight at 4 °C. The cells were then washed and incubated for 90 min with the fluorescent secondary antibody. The cells were washed again, and DAPI probe (1 µg/mL, (Santa Cruz Biotechnology, Dallas, TX, USA)) was added for 5 min. A ZEISS LMS 700 confocal microscope (Carl Zeiss, Oberkochen, Germany) was used for image acquisition and digital image recording.

### 2.4. Doubling Time of Primary Cells

The growth of cells was quantified using the IncuCyte® imaging device (Sartorius Stedim Biotech, Göttingen, Germany). A picture of the cells was taken every 3 h until confluence. Five thousand cells were plated in a 96-well plate in culture medium 2 and incubated at 37 °C. The culture medium was changed at 24 h and the plate positioned in the IncuCyte® incubator. The culture medium was changed every 2 to 3 days. Growth versus time curves and doubling times were performed with IncuCyte software (Sartorius Stedim Biotech, Göttingen, Germany).

### 2.5. lEV Isolation

Three types of lEVs were isolated: circulating lEVs (cEV) isolated from blood of patients before nephrectomy, lEVs from the supernatant (sEV) of the primary culture of cancer cells, and control lEVs (iEV) isolated from the blood of individuals with no medical history or cancer.

cEV were isolated from a fasting blood sample collected in the morning or the day before surgery. Five EDTA tubes (Vacutainers, Becton Dickinson, Le Pont de Claix, France) were collected and centrifuged 4 times within two hours of collection (15 min at $260\times g$ and 20 min at $1500\times g$ to remove blood cells, and the remaining plasma was centrifugated for 45 min at $17,000\times g$ to pellet lEVs and then for another 45 min at $17,000\times g$ again to wash and concentrate the cEV). The cEV were then re-suspended in 200 µL of saline (0.9% NaCl) and stored at +4 °C. All steps were performed in a sterile environment. iEV were isolated using the same protocol.

sEV were isolated from the culture medium of the primary cells. The supernatant from cells cultured in a T175 $cm^2$ flask was centrifuged after 48–72 h of incubation until achieving a confluence close to 80–90%. A first centrifugation at $300\times g$ for 10 min at 0 deceleration was performed. The resulting supernatant was centrifuged again at $2000\times g$ for 10 min. A centrifugation at $20,000\times g$ for 3 min was performed to eliminate the last cell fragments. Finally, two successive centrifugations at $13,000\times g$, each for 45 min at

4 °C, were performed to pellet and wash the sEV. The resulting pellet was stored at 4 °C in 200 μL of 0.9% NaCl.

The protein concentration of lEVs samples was measured using the Lowry method with BSA as standard on a 96-well plate and was measured with a spectrophotometer (Victor2, (PerkinElmer, Waltham, MA, USA)) at the wavelength of 570 nm [22].

cEV and iEV were previously characterized in the manuscript Vergori et al. [23].

### 2.6. Animal Model

All animal experiments were performed with approved institutional agreement (APAFI S#28858-2021010611224905 v6) and received a favorable opinion from the Pays de la Loire animal experimentation ethics committee no. 006. Twenty 6-week-old male nude balb/c mice (Charles River, L'Arbresle, France) were randomly divided into 5 groups (n = 4 for each group). Each group received the xenograft and the lEVs from the same patient into the right flank. In each group, the first mouse received the patients' cEV, the second mouse received the associated primary culture sEV, the third received the iEV (from individual without cancer), and the fourth received an equivalent volume of NaCl as control. Mice received a total of 5 subcutaneous injections of 100 μL of lEVs at 200 mg/mL. Injections were administered 3 and 1 day before the injection of the xenograft. Then, at day 0, 100 μL of lEVs was injected with $4.25 \times 10^7$ cells in 100 μL of matrigel (Corning Matrigel High Concentration, Corning, Life science, Amsterdam, The Netherlands) for a total volume of 200 μL and then at 7 and 14 days after the xenograft implantation.

After nine weeks, tumor size was measured by ultrasound echography (Vevo 770, Fujifilm VisualSonics, Toronto, ON, Canada) and was calculated using the formula: $v = L \times l^2 \times 0.5$, where "L" indicates the large diameter and "l" indicates the small diameter as previously described [24]. The kidneys and xenografts were harvested and frozen after sacrifice.

### 2.7. Immunofluorescent Labeling of Tumors

Frozen tumor sections (10 μm thickness) were prepared using a Leica CM3050 S cryostat (Leica Biosystems, Wetzlar, Germany). Different tumor sections were incubated with anti-CD31 or anti-Ki67 antibodies (Cell Signaling Technology, Danvers, MA, USA). After washing, bound antibodies were detected with a secondary peroxidase-conjugated anti-mouse antibody (Santa Cruz Biotechnology). Nuclei were counterstained with DAPI (Santa Cruz Biotechnology). Confocal microscopy (ZEISS LMS 700) and digital image recording were used. Images were analyzed using ImageJ software (National Institutes of Health, Bethesda, MD, USA).

### 2.8. Western Blot Analysis of Mouse Native Kidney

Kidneys were homogenized and lysed. Protein extracts (80 μg per lane) were separated on polyacrylamide gel by electrophoresis and then transferred to nitrocellulose membrane (Bio-Rad, Hercules, CA, USA). Immuno-blotting was performed by incubating each membrane with anti-MMP2 (Santa Cruz, Dallas, TX, USA), anti-Ca9 (Cell Signaling Technology) and anti-ß-actine (Sigma-Aldrich). After washing, bound antibodies were detected with a secondary peroxidase-conjugated anti-rabbit antibody (Sigma-Aldrich), anti-goat (Sigma-Aldrich) or anti-mouse antibody (U.S. Biological, Salem, MA, USA). The chemiluminescent revelation was performed by a detection kit with luminol (Santa Cruz). The protein–antibody complexes were detected by ImageQuant™ LAS 400 (GE Healthcare, Chicago, IL, USA). Quantification was performed by densitometric analysis using ImageJ software and normalized to actin.

### 2.9. Statistical Analysis

Results are expressed as mean ± standard deviation (SD) of n independent observations. When their distributions were normal, the data were then analyzed by an ANOVA test; otherwise, a Kruskal–Wallis test was used. All statistical analyses were performed

using Prism software (v.8.00; GraphPad Software, La Jolla, CA, USA). Differences were considered significant when $p < 0.05$ regardless of the statistical test used.

## 3. Results

### 3.1. Primary Cell Line Characterization

To create our study model, we established five primary cell lines from five different patient nephrectomy specimens operated for cancer. Patients included were two females and three males. Median age of the patients was 63.8 years (SD 12.4). According to the American Joint Committee on Cancer (AJCC) Tumor Node Metastasis (TNM) classification, there was one T1, two T2 and two T3 tumors. Only one patient had metastasis. There was no lymph node involvement. The median tumor size was 8.6 cm (SD 1.7). Regarding the aggressiveness of the patients' tumor cells, according to the International Society of Urological Pathology (ISUP) score, which ranges from 1 to 4, there were two ISUP 2 scores, two ISUP 3 scores and one ISUP 4 score. Two patients had tumor necrosis on the pathology specimen, which is a poor prognostic factor. (Table 1).

Cell growth curves (Figure 1A) and doubling time determination were performed using IncuCyte®. The average doubling time was 30.4 h (±9.1 h), which was variable depending on the patient (ranged from 22 h to 40 h).

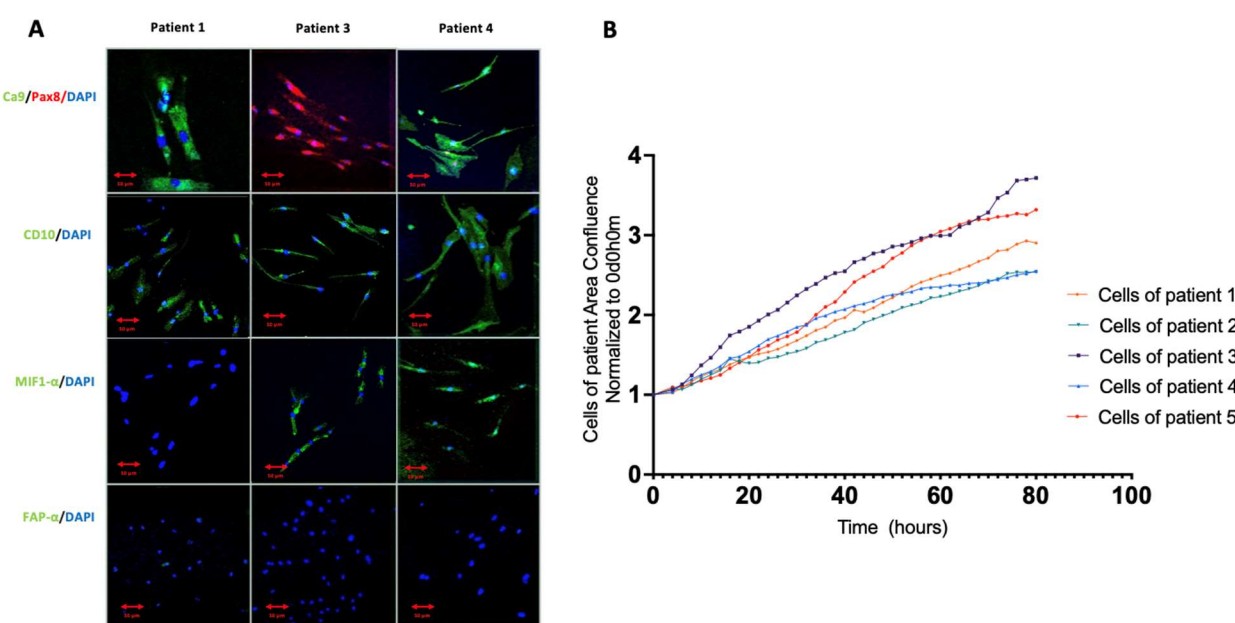

**Figure 1.** Characteristics and growth of primary culture cell line. Cells isolated from the patients have a renal and cancerous phenotype. Their proliferation was stable and continuous over time. (**A**) Confocal microscopy images of cells from patients 1, 3 and 4. (**B**) Proliferation of all primary culture lines as a function of time. Confluence at T0 is normalized to 1 on the graph and then measured as a function of time by the Incucyte ® live Cell Analysis software.

Immunofluorescence analysis of cells used for xenografts confirmed their renal and cancerous origins (Figure 1B). Markers of kidney cancer commonly used in the pathology laboratory or in in vitro studies were sought on our cells, such as carbonic anhydrase 9, neprilysin (CD10) and PAX 8 [25–27]. Macrophage migration inhibitory factor (MIF1 alpha) expression was also measured as a marker for kidney cancer [28]. Finally, fibroblast activating protein alpha (FAP-alpha) was sought as a marker of potentially contaminating fibroblasts in the primary culture [29].

Cells from all patients expressed Ca9 and CD10, except for the cells from patient 3, which did not express Ca9. In contrast, cells from patients 3 and 4 expressed Pax8 on their surface. MIF1 alpha was expressed only by cells from patient 3. All cells expressed CD10.

Finally, FAP-alpha labeling was slight (only few cells from patient 2 expressed FAP-alpha, not shown) excluding contamination by fibroblasts.

### 3.2. Effect of lEVs on Tumor Growth and the Peritumoral Environment

### 3.2.1. lEVs from Cancer Patients Tend to Increase Xenograft Growth

Each of the five primary cells were injected into the flank of a group of four mice. Thus, there were five groups composed of four mice that had the xenograft from the same patient. The first mouse in the group received the lEVs isolated from the patients' bloodstream (cEV), the second received the lEVs isolated from the supernatant of the primary culture of the xenograft (sEV), the third received the lEVs from the bloodstream of individuals without disease or cancer (iEV), and the last received an injection of saline solution as the control group (NaCl). The injection protocol is detailed in Section 2.

Ultrasound analysis of the xenograft after 9 weeks of growth showed a small subcutaneous nodule in all mice. The nodule was visible to the naked eye in 14 out of 20 mice. The xenografts had a mean volume of 7.733 mm$^3$ (SD: 7.7). No significant difference was observed between the xenograft volumes of the different groups of mice. However, there was a clear trend. The mean volume of xenografts treated with either cEV (mean: 14.21 mm$^3$, SD: 11.2) or sEV (mean: 10.41 mm$^3$, SD: 8) was greater than the mean volume of xenografts in the group treated with iEV (mean: 7.7 mm$^3$, SD: 7.7) and NaCl control group (mean: 1.6 mm$^3$, SD: 1.4) (Figure 2). During the ultrasound examination and dissection of the mice, organs such as lungs, liver, kidneys and spleen were examined. We did not observe any signs of visible metastasis of the xenograft.

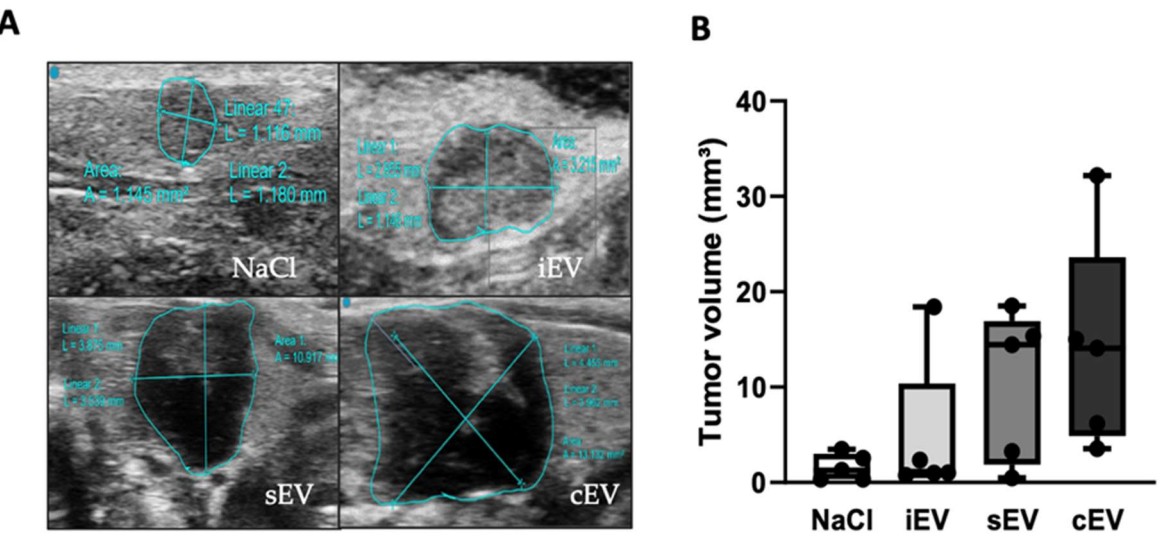

**Figure 2.** Measurement and comparison of xenograft volume. lEVs isolated from the patients' bloodstream (cEV) and from the primary culture supernatant (sEV) altered the peritumoral environment. (**A**) Representative images of xenograft volumes measured by ultrasound in the different groups after 9 weeks of growth. (**B**) Box plot of tumor volume in the different groups after 9 weeks of growth. Results are expressed in cubic millimeters (mm$^3$). The analysis was performed by a Kruskal–Wallis test on GraphPad Prism8 software.

### 3.2.2. lEVs from Cancer Patients Altered Peritumoral Environment

Analysis of the tumor environment by confocal microscopy revealed a statistically significant increase in CD31 (platelet endothelial cell adhesion molecule-1, PECAM-1) expression in the xenografts treated with cEV (mean: 9.1 ± 3.5) compared to the NaCl control group (mean: 5.1 ± 1.1) ($p = 0.0378$) (Figure 3). There also appeared to be a tendency for the sEV group (mean: 7.4 ± 1.9) to overexpress CD31 compared to the NaCl control group and iEV group (mean 5.8 ± 0.99) (Figure 3).

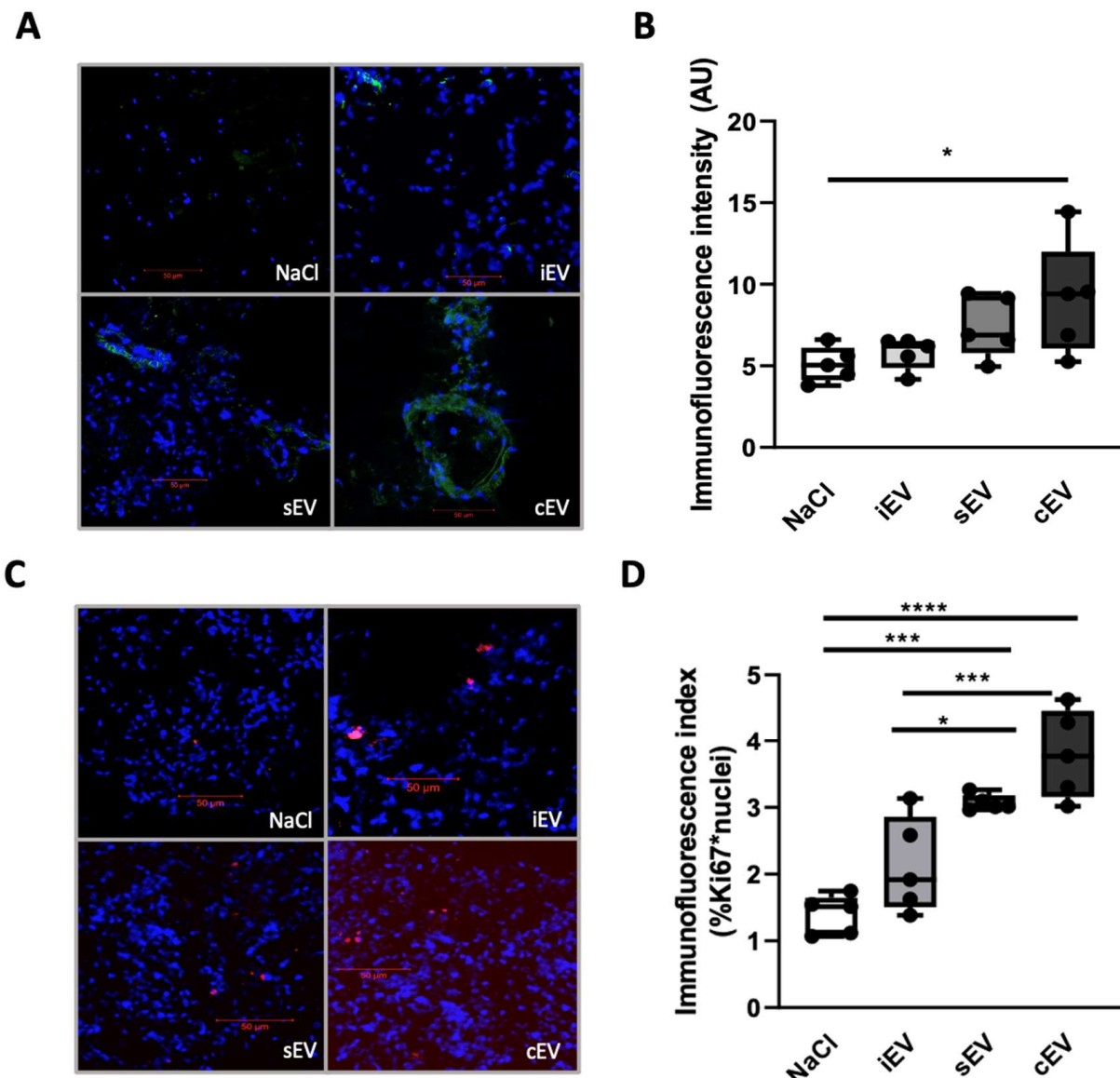

**Figure 3.** Evaluation of CD31 and Ki67 index in xenograft by confocal microscopy (objective ×20). (**A**) Representative images of CD31 labeling of the xenograft. Blue fluorescence represents DAPI (nuclei), and green fluorescence represents endothelial marker CD31 labeling. (**B**) Quantification: results are expressed in arbitrary immunofluorescence units (AU). (**C**) Representative images of nuclear labeling of xenograft by Ki67. Blue fluorescence represents DAPI (nuclei), and red fluorescence represents the marker of proliferation Ki-67. (**D**) Quantification: results are expressed as immunofluorescence index per nucleus (%Ki67 + nuclei). Scale bar = 50 μm. Data processing coupled with an ANOVA test was performed on GraphPad Prism8 software. Data are expressed with a box plot, (n = 5), * $p < 0.05$, *** $p < 0.001$, **** $p < 0.0001$.

Regarding the proliferation marker Ki67 index, there was a statistically significant increase in the cEV group (mean: $3.8 \pm 0.67$) compared to the NaCl control group (mean: $1.4 \pm 0.3$) ($p < 0.0001$) and the iEV group (mean: $2.1 \pm 0.72$) ($p = 0.005$). The sEV group (mean: $3.1 \pm 0.12$) also had a higher Ki67 index than the NaCl control group ($p = 0.0005$) and the iEV group ($p = 0.0482$) (Figure 4).

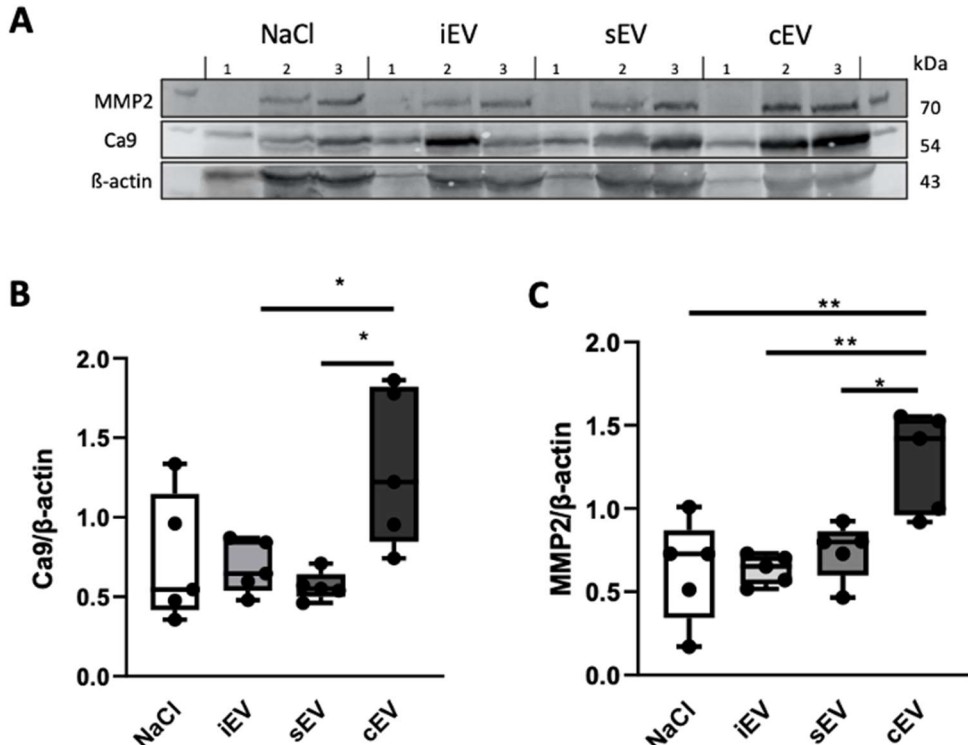

**Figure 4.** Western blot analysis of Ca9 and MMP2 expression in mouse kidney. lEVs isolated from the bloodstream of patients (cEV) altered the native mouse kidney. Quantification and comparison of Ca9 and MMP2 proteins. Results are expressed as Ca9/β-Actin ratio. (**A**) Representative Western blot images of kidneys from mice 1, 2 and 3 of 5 in different groups. Line 1 MMP2: 70 kDa; line 2 Ca9: 54 kDa; line 3 β-Actin: 43 kDa. First and last lines correspond to protein ladders (Thermo Fisher Scientific, Waltham, MA, USA). (**B**) Ca9/β–actin and (**C**) MMP2/β–actin ratios (n = 5). Data processing, coupled with an ANOVA test, was performed on GraphPad Prism8 software. Data are expressed with a box plot (n = 5), * *p* < 0.05, ** *p* < 0.01.

These results show that lEVs from cancer patients, that is cEV and sEV, favor vascularization of the peritumoral environment and increase tumor cell proliferation3.2.3. cEV Affects Remote Organs from the Xenograft

After the mice had been sacrificed, organs other than the xenograft were extracted for analysis. We wanted to examine whether the lEVs injected into the mice had caused tissue alteration at a distance from the injection site and at a distance from the xenograft. Expression of the hypoxia marker carbonic anhydrase 9 (Ca9) and tissue remodeling matrix metalloproteinase-2 (MMP2) in the native mouse kidney was measured by Western blot. Ca9 expression was significantly increased in the cEV-treated group (mean: $1.31 \pm 0.49$) compared to the sEV-treated group (mean: $0.57 \pm 0.09$) ($p = 0.0133$) and the iEV-treated group (mean: $0.69 \pm 0.17$) ($p = 0.041$). sEV and iEV had no effect on Ca9 expression compared to the saline group (mean: $0.73 \pm 0.40$). MMP2 expression was also increased in the cEV group (mean: $1.28 \pm 0.3$) compared to the NaCl control group (mean: $0.63 \pm 0.31$) ($p = 0.0025$), iEV (mean: $0.63 \pm 0.09$) ($p = 0.026$) and sEV (mean: $0.74 \pm 0.17$) ($p = 0.0116$) groups (Figure 4).

## 4. Discussion

lEVs are an important mechanism of intercellular communication. Several studies have highlighted their involvement in the development and progression of several cancers. However, these studies have mostly evaluated lEVs produced in vitro by cells from commercial lines. In the present study, we developed an original model to evaluate the role of lEVs from renal cancer patients on their own cancer cells and the peritumoral environment.

We demonstrated that cEV isolated from the blood of patients before nephrectomy and sEV isolated from the supernatant of the primary culture had a positive effect on the xenograft. This result is related to the tendency to support the development of xenograft volume, increased vascularization and tumor cell proliferation. The cEV also altered organs distant from the xenograft, such as the primitive kidney tissue of the mouse.

Regarding the establishment of the primary culture, we took our lead from previous studies [30–33]. We selected patients with the largest tumors and the highest ISUP scores to optimize the in vitro survival of the primary culture and the xenograft [34]. With the culture medium used, our cells were affected by neither bacterial contamination nor growth arrest. Characterization by immunofluorescence allowed us to confirm their renal and cancerous origins but also to eliminate fibroblast contamination. All primary cultures showed kidney cancer markers such as CA9, Pax8, CD10 or MIF1a [25–28]. One primary line obtained had some cells that expressed the marker for activated fibroblasts (FAP). In the literature, FAP is expressed by kidney cancer stromal fibroblasts and is associated with tumor aggressiveness [29]. We probably had a slight contamination of the primary cancer cells of this patient by some cancer-associated fibroblasts. Since the number of PAF-expressing cells was low, we did not consider this result as abnormal and kept the cell line for study. The measurement of the doubling time of our cells with Incucyte® bears important information. The doubling time of the different cell lines can vary from simple to double (22 to 40 h). To compare the commercial lines widely used in vitro, Lobo et al. (2016) reported a doubling time of 16 h for 786-0 cells, which is one of the most widely used commercial renal cell carcinoma cell lines [31].

lEVs isolated from the patients' bloodstream (cEV) and the supernatant of the primary sEV culture showed a positive trend on xenograft volume. However, we could not find a statistically significant difference. It is plausible that increasing the time to tumor growth or the number of mice may have been able to achieve a significant result. We also questioned the low volume of the xenografts. The initial protocol called for the mice to be sacrificed at 7 weeks after xenograft injection in accordance with the protocol of previous studies. We extended it to 9 weeks to allow for time for the xenograft to grow. The number of cells injected seems to be relevant in relation to the data in the literature [17,19,20,35–37]. Verhoest's (2014) thesis shows that the injection of human umbilical vein endothelial cells (HUVEC) with primary kidney cancer cell lines from patients [38] promotes vascularization of the xenografts and increases their volume. This xenograft technique is also used in the study of Yorozu et al. (2020) in which they show that injection of HUVEC with commercial DLD1 colorectal cancer cells increased the volume and vascularization of the xenograft in a mouse model [39]. It would be interesting to perform further mouse series by adding this step to our protocol.

CD31 labeling by confocal microscopy illustrates a significant increase in xenograft vascularization in cEV-treated mice compared with the NaCl control group. However, there is only a trend toward increased xenograft vascularization in mice treated with sEV. Ki67 markings showed that both cEV and sEV promote cell entry into mitosis. These results suggest that lEVs had a positive effect on tumor growth by promoting vascularization and cell proliferation. These in vivo results confirm the in vitro results of previous studies. Zhang et al. (2013) and Grange et al. (2011) reported that lEVs derived from renal cell carcinoma cells increased angiogenesis in vivo and had a positive effect on vascular endothelial growth factor (VEGF) expression [19,20]. A study by Elham Hosseini-Beheshti et al. (2016) demonstrated that lEVs of commercial prostate cancer cell lines promoted the in vitro proliferation of LNCaP prostate cancer cells but also increased the volume and Ki67 expression in a mouse model [40]. Nevertheless, our study confirms these results from the literature in a study model where the cells and lEVs are obtained from human samples and from the same person.

Furthermore, circulating lEVs isolated from patients (cEV) resulted in organ modification distant from the xenograft. lEVs injected subcutaneously in contact with the xenograft probably migrated via the bloodstream into the native mouse kidney. Western blot analysis

of native mouse kidney tissue shows that cEV increased MMP2 expression in the kidney. In the literature, MMP2 is a matrix metalloproteinase associated with cancer cell invasion and metastasis of many cancers including kidney cancer [41]. lEVs from cancer cells are known to increase tissue remodeling and MMP2 expression [42]. In addition, in the native mouse kidney, we found an increase in Ca9 expression by cEV. Ca9 is described as a hypoxia-related protein that is overexpressed by clear cell renal cell carcinoma cells [43–45]. This result shows tissue damage and possible dissemination of cancer cells induced by the patients' lEVs. Nevertheless, we did not observe any macroscopic metastasis during dissection of the animals or on ultrasound examination before sacrifice. Tumor lEVs are known to convert normal fibroblasts into tumor-associated fibroblasts, a process that also leads to the formation of metastatic niches and promotes cancer metastasis [46,47]. It is possible that lEVs from cancer patients generate a pre-metastatic niche with a few Ca9-expressing cancer cells and stromal cells with a cancerous phenotype over expressing MMP2. In the same context, Grange et al. (2011) reported that the treatment of SCID mice with lEVs from kidney cancer cells significantly increased the number of lung metastases induced by intravenous injection of renal carcinoma cells [20]. These results suggest that the organs of future metastasis are not passive receptors for circulating tumor cells. The primary tumor actively modifies distant organs. lEVs are one of the mechanisms involved in the modification of organs at a distance from the primary cancer to form a metastatic niche and to transform localized cancer into metastatic disease.

One of the limitations of the study is that we only characterized a portion of the lEVs that we injected. The protocol for extraction of cEV and iEV and their characterization have already been discussed by Vergori et al. (2021) [23]. However, no assay method other than the method of Lowry (1951) has been performed on sEV. This represents an important limitation of the study. Furthermore, it is likely that the lEVs we studied are composed of heterogeneous populations with different membrane or cytosolic characteristics. We hypothesize that there is a specific subpopulation of circulating lEVs that would be responsible for the observed effects or, on the contrary, if among all the injected lEVs, some had a negative effect on the cancer. It would be interesting, in a future study, to characterize all the lEVs used to define subpopulations according to their surface protein or cytosol content. For instance, it is known that kidney cancer cells overexpress Ca9 and that kidney cancer patients have higher levels of Ca9-overexpressing lEVs than patients without cancer [23,44]. Horie et al. (2017) showed that exosomes carrying Ca9 promoted neo-angiogenesis and MMP2 expression on HUVEC cells [48]. Whether the effects described in the present study are associated with cEV-carrying Ca9 remain to be further studied. Finally, we performed limited elucidation of the molecular mechanism by which lEVs increase xenograft size or alter the peritumoral environment. Studies with a finer characterization of lEVs and a better understanding of the involved molecular mechanism must be carried out.

Currently, there are no therapies in current practice that aim to limit the effect of EV on cancer progression. However, reducing the impact of cancer-derived lEVs could be a novel strategy for treating cancer patients. Three potential strategies have been proposed: decreasing lEV production, eliminating circulating lEVs and inhibiting lEV absorption [49,50]. Pre-clinical studies have already shown very interesting results. Kosaka et al. (2013) showed that inactivation of neutral sphingomyelinase 2 (nSMase2), which is a protein involved in lEV release, led to a reduction in miRNA secretion in EVs and inhibited lung metastasis of breast cancer in a mouse model [51]. In addition, Nishida-Aoki et al. (2017) reported that administration of CD9 and CD63 antibodies increased lEV clearance via macrophages and decreased metastasis in a breast cancer xenograft model [52]. These strategies seem promising as means to prevent metastasis in some cancers. However, it is necessary to use a mechanism specific to the cancer cells in each case so as not to alter the functioning of healthy cells. To date, there does not seem to be a therapeutic strategy that can be used in humans.

## 5. Conclusions

In conclusion, the lEVs we studied had a positive effect on the patients' cancer development in our mouse model. Although we found only a positive trend on the volume of the xenograft, the lEVs significantly increased vascularity and cell proliferation. In addition, these lEVs resulted in alterations in mouse organs distant from the injection site. These results suggest that lEVs from cancer patients are involved in the development and progression of their own tumors. lEVs may also support the development of a pre-metastatic niche distant from the primary cancer and promote the development of the cancer to a metastatic stage. In the future, we may consider a preclinical study to block the tumorigenic action of lEVs. The protocol would consist of identifying a population of highly tumorigenic lEVs responsible for the observed effect. The characterization of this population could allow for the identification of a specific surface protein. A therapy specifically targeted against this surface protein would allow for a degradation of tumorigenic lEVs by macrophages without altering the functioning of the organism's healthy cells.

**Author Contributions:** M.F., P.B. and M.d.C.M. designed the study. M.F. wrote this article. L.V. supervised this study. S.B. was responsible for anatomopathological part. V.L.C. was responsible for the patient inclusion part. All authors have read and agreed to the published version of the manuscript.

**Funding:** This work was supported by a Grant from the French Association of Urology and the LES AILES foundation. The APC was funded by "association Angevine d'études urologiques".

**Institutional Review Board Statement:** The animal study protocol was approved by the Institutional Ethics Committee of the Pays de la Loire animal experimentation ethics committee no. 006. the agreement of the French Ministry of "Higher Education, Research and Innovation" for animal research has been obtained and is referenced under the number APAFIS#28858-2021010611224905 v6.

**Informed Consent Statement:** All patients in this study were from the national database "UroCCR" (NCT03293563, CNIL authorization number: DR-2013-206). Patients were informed about the purpose of the study, and consent was obtained in writing.

**Data Availability Statement:** No new data were created or analyzed in this study. Data sharing is not applicable to this article.

**Acknowledgments:** We would like to thank the "Service Commun des Animaux de l'Hôpital Universitaire" (SCAHU) of Angers and the SCIAM technical platform of the University of Angers (SFR4208) for their hospitality and cooperation in this study.

**Conflicts of Interest:** The authors declare no conflict of interest.

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
