# Peer review of "Effects of Large Extracellular Vesicles from Kidney Cancer Patients on the Growth and Environment of Renal Cell Carcinoma Xenografts in a Mouse Model"

_cimb, doi:10.3390/cimb45030163_

Round 1

Reviewer 1 Report

1.     This manuscript is interesting and well-done.

2.     The strength of this article is well organized for readers to understand for issue of lEVs from cancer patients is involved in their own tumor development and progression.

3.     But the regretting point is wishing it was more attentive about showing STDEV is too broad in several figure. Fig. 2B (iEV, cEV), Fig.3B (cEV), Fig. 4B (Nacl) and C(Nacl, sEV). It's just my opinion, standard deviation seems a little high. Could you tell me reason of high STDEV? If there is a large difference in the STDEV, the reliability of the data can be reduced. This point must be taken very seriously.

4.     Figure 3 A and C, the scale bar is not clearly visible. It would be better to adjust the resolution so that the scale bar can be clearly seen.

5.     Present stidy described ‘lEVs are submembrane fragments released into the extracellular environment and involved in intercellular communication [9]. Their biogenesis results from a contraction phenomenon of the cell cytoskeleton after an increase in cytoplasmic calcium [10,11].’ in introduction. Increase of the cytoplasmic free calcium is known as induction of apoptosis. What is the relationship between lEVs and apotosis?

6.     It's just my opinion, this manuscript was accepted to be enough after english language and style are minor spell check.

Author Response

Cover letter:

Thank you very much for allowing us to resubmit. We have carefully considered the comments provided by all reviewers. We thank them for their insightful comments. Our responses to each of the reviewers' comments are provided in a Word document. We have had the Manuscript proofread by a competent service for English grammar correction. Please find enclosed the certificate.

We thank you once again for your consideration of this article.

Yours sincerely,

Pierre Bigot

Reviewer 2 Report

Large extracellular vesicles (lEVs) are implicated in several pathophysiological situations including cancer. However, to date, no studies have evaluated the effects of lEVs isolated from patients with renal cancer on the development of their own tumor. In this study, the authors investigated the effects of three types of lEVs on the growth and peritumoral environment of xenograft clear cell renal cell carcinoma in a mouse model. They made the conclusions that lEVs from patients with cancer are involved in their own tumor growth and progression. However, the overall experimental designs are too rough. The results demonstrated in this study did not support the conclusion. My decision is rejection.

1.      The resolution of figure 1 is too low.

2.   The authors should provide more detail data to identify the primary culture cells. Ca9 or CD10 is expressed in both cancer cells and fibroblasts. They are not specific markers for kidney cancer cells. The primary culture cells from the surgical specimen could be cancer cells or fibroblasts. Therefore, the results in this study are not reliable.

3.     What is the clinical and molecular relevance of Ca9 and MMP2 expression in the mouse kidney after IEV treatment?

4.     No characteristic data of IEVs obtained from pre-nephrectomy patient blood or supernatant of primary cancer cell culture or blood from individuals with no medical history were demonstrated in this study.

5.      The molecular mechanisms evaluated in this study are too superficial.

Author Response

(The authors gave the same response as above.)

Reviewer 3 Report

In this study,  the authors have investigated the effects of plasma membrane-derived vesicles large extracellular vesicles (lEVs)  isolated from patients with renal cancer on the growth of xenograft mouse model of clear cell renal cell carcinoma. IEVs increased the volume of the mouse xenografts and they further found higher levels of the proliferating marker Ki67 as compared to control Saline group. They also found that MMP2 expression was increased in the IEVs treated group.

This study is novel and can provide new insights in the treatment of renal cell carcinoma. 

However, the limitation of the paper is that the author have not characterized the IEVs. 

Author Response

We thank the reviewer for studying our manuscript and for his insightful comments.

Yours sincerely, 
Pierre Bigot 

Round 2

Reviewer 2 Report

The authors have improved the manuscript somewhat. However, I did not find any necessary changes in the manuscript that would answer the main questions raised in my last review.

Author Response

Dear Reviewer,

We thank you for giving us the time to modify Figure 1 and allow us to resubmit our application. We have made the changes in the manuscript based on your feedback. Our responses and summary of manuscript changes are provided in a Word document.

We thank you once again for your consideration of this article.

Yours sincerely,

Pierre Bigot
